# IFNγ-Treated Macrophages Induce EMT through the WNT Pathway: Relevance in Crohn’s Disease

**DOI:** 10.3390/biomedicines10051093

**Published:** 2022-05-08

**Authors:** Dulce C. Macias-Ceja, Sandra Coll, Cristina Bauset, Marta Seco-Cervera, Laura Gisbert-Ferrándiz, Francisco Navarro, Jesus Cosin-Roger, Sara Calatayud, María D. Barrachina, Dolores Ortiz-Masia

**Affiliations:** 1CIBERehd (Centro de Investigaciones en Red Enfermedad Hepática y Digestiva), Departamento de Farmacología, Facultad de Medicina, Universidad de Valencia, 46010 Valencia, Spain; macias.dcc@gmail.com (D.C.M.-C.); sandra.coll@uv.es (S.C.); cristina.bauset@uv.es (C.B.); laura.gisbert@uv.es (L.G.-F.); sara.calatayud@uv.es (S.C.); dolores.barrachina@uv.es (M.D.B.); 2FISABIO (Fundación para el Fomento de la Investigación Sanitaria y Biomédica), Hospital Dr. Peset, 46017 Valencia, Spain; marta.seco@uv.es; 3Servicio Cirugía y Coloproctología, Hospital de Manises, 46940 Valencia, Spain; fran.navarro.vicente@gmail.com; 4Departamento de Medicina, Facultad de Medicina, Universidad de Valencia, 46010 Valencia, Spain

**Keywords:** Crohn disease, IFNγ, Epithelial Mesenchymal Transition (EMT), macrophages, fibrosis

## Abstract

Background: Fibrosis is a common complication of Crohn’s disease (CD) in which macrophages play a central role. Epithelial-mesenchymal transition (EMT) and the WNT pathway have been associated with fibrosis. We aim to analyse the relevance of the tissue microenvironment in macrophage phenotype and the EMT process. Methods: Intestinal surgical resections are obtained from control and CD patients with stenotic or penetrating behaviour. Cytokine’s expression, macrophage phenotype, EMT markers and WNT signalling pathway are determined by WB, RT-PCR, ELISA or Cytometry. U937 cells are treated with IFNγ, TNFα, IL1β, IL4 or IL10 and co-cultured with HT29 cells and, in some cases, are treated with XAV939 or *miFZD4*. The expression of macrophage, EMT and WNT pathway markers in U937 or HT29 cells is analysed by WB or RT-PCR. Results: IFNγ, WNT6, CD16 and CD86 are increased in the intestinal tissue of CD patients. IFNγ-treated U937 activated the EMT process and WNT pathway in HT29 cells, and the EMT process is mediated by FZD4. Conclusions: An IFNγ-rich microenvironment polarises macrophages, which induces EMT through the WNT pathway.

## 1. Introduction

Fibrosis represents a major challenge in Crohn’s disease (CD). Around 50% of CD patients will develop fibrotic strictures (B2 behaviour) or penetrating lesions (B3 behaviour), and up to 75% will eventually need surgery [1]. The mechanism underlying the development and progression of inflammatory bowel disease (IBD) is unclear, but growing evidence suggests that immunological lesions are produced by the infiltration of Th1 cells, which produce inflammatory cytokines [2]. IFNγ is a critical mediator of inflammatory disorders and has been implicated in the pathology of IBD [3,4].

In recent years, the relevance of epithelial-mesenchymal transition (EMT), a cellular trans-differentiation programme by which epithelial cells acquire mesenchymal features, has become apparent in CD complications [5,6]. The WNT signalling pathway plays a fundamental role in EMT, and we have recently reported increased EMT associated with enhanced WNT2b/FZD4 interaction in intestinal tissue from CD patients with penetrating behaviour, thus demonstrating that WNT2b induces EMT through the activation of FZD4 [7]. However, the source of this enhanced WNT2b is yet to be established.

Macrophages are immune cells implicated in the innate immune response, and they perform different functions, including the inflammatory process, wound healing and fibrosis [8,9]. The different functions arise from the way macrophages change their phenotype and release a variety of biological mediators depending on their surroundings. Macrophages have been shown to be a significant source of WNT ligands in adult tissues [10,11,12]. In fact, evidence demonstrates that, depending on their phenotype, macrophages secrete different kinds of ligands involved in the WNT pathway [12,13]. These observations, together with the fact that these cells usually accumulate when there is chronic inflammation, point to the relevance of macrophage-derived WNT ligands in mucosal regeneration and in complications such as fibrosis and cancer. However, we have recently reported that the WNT2b ligand is expressed mainly in epithelial cells and not so much in the lamina propria of the intestine [7]. All these observations have led us to analyse the macrophage phenotype that is mainly expressed in the intestine of CD patients with stenotic or penetrating complications and to determine the role that these cells play in the regulation of the WNT signalling pathway and, in turn, in EMT.

To do this, we explored the relevance of the tissue microenvironment for macrophage phenotype and the EMT process through the WNT pathway and their implication in the complications of CD.

## 2. Materials and Methods

### 2.1. Patients

Lennard-Jones’ criteria (clinical, biological, endoscopic and histological) were used for the diagnosis of CD. Patients with severe refractory CD who had ceased to respond to second-line treatment and required surgery were classified with stricturing (Montreal classification: B2) or penetrating (Montreal classification: B3) behaviour. Digestive surgeons obtained terminal ileum samples from the stricture in the B2-CD patients or from the damaged area of intestine in the patients with penetrating behaviour (B3-CD patients). In all cases, patients had received biological therapy (anti-TNFα) three weeks before surgery. Surgical specimens from unaffected mucosa from the terminal ileum of right colon cancer patients were used as controls (non-IBD). Non-IBD patients were not undergoing chemotherapy before or at the time of surgery. Information concerning all the patients analysed in this study can be consulted in Table 1.

### 2.2. Cell Culture and Treatments

U937-monocytes (European Collection of Cell Culture, Salisbury, UK) were cultivated in RPMI medium (supplemented with the following: 10% inactivated foetal bovine serum (FBS); 100 U/mL penicillin; 100 μg/mL streptomycin). U937-monocytes were segregated into macrophages and RPMI medium phorbol myristate acetate (PMA) was added for 48 h [14]. U937-derived macrophages were stimulated with interferon γ (IFNγ; 10 ng/mL), interleukin 4 (IL4; 20 ng/mL), interleukin 10 (IL10; 10 ng/mL), tumor necrosis factor α (TNFα, 10 ng/mL) or interleukin 1β (IL1β; 10 ng/mL) for 96 h.

Human colonic epithelial cells (HT29, American Type Culture Collection, VA, USA) were cultured in McCoy’s (Modified) Medium (Sigma-Merck, Madrid, Spain) supplemented with 10% inactivated FBS, 100 U/mL penicillin, 100 μg/mL streptomycin and 2 mM L-glutamine. HT29 cells were co-cultured with U937 macrophages using Transwell inserts (Corning Incorporated, Madrid, Spain) with a 0.4 mm porous membrane [15]. U937-derived macrophages were seeded on the inserts and treated with IFNγ, IL-4, IL10 or IL-1β for 96 h. Subsequently, the inserts were placed on top of HT29 cells and maintained in co-culture for 72 h. When appropriate, HT29 cells were treated with the inhibitor of the WNT-pathway, XAV939 1 μM, (Thermo Fisher Scientific, Valencia, Spain) during co-culture with U937 cells [12].

Alternatively, HT29 cells were transfected with the following vector-targeting human FZD4 (*miFZD4*, targeting sequences: 5′-CGGCATGTGTCTTTCAGTCAA-3′ and 5′-GTCACTCTGTGGGAACCAATT-3′ (GenBank Accession No. NM_012193), as described previously [7]. Twenty-four hours post-transfection, the cells were co-cultured with IFNγ-treated U937 cells for 72 h.

### 2.3. Isolation of Macrophages and Secretomes from Intestinal Mucosa

Lamina propria mononuclear cells (LPMCs) were obtained from surgical samples by digestion, as previously described [16]. The LPMCs were analysed by flow cytometry.

Surgical samples from CD and non-IBD patients were cut into 10 mm pieces and incubated for 15 h in RPMI medium supplemented with 10% inactivated FBS with 100 U/mL penicillin and 100 μg/mL streptomycin. Subsequently, the supernatant was centrifuged for 5 minutes at 1500 rpm and used as a secretome.

### 2.4. Differentiation of Monocytes and Secretome Treatment

Ficoll density gradient (400 g for 40 min) was used to isolate human peripheral blood mononuclear cells from healthy donors. The monocyte-derived macrophages (MDMs) were seeded in 12-well culture plates and then differentiated into macrophages by culturing half with ImmunoCultTM –SF Macrophage Differentiation Medium (STEMCELL technology, Cambridge, UK) and half with the secretome (non-IBD, B2 or B3) supplemented with 50 ng/mL recombinant human M-CSF (Peprotech, Hamburg, Germany) at 37 °C in 5% CO_2_ for 6 days (the medium was renewed on day 3).

### 2.5. IFNγ ELISA

Secreted protein levels of IFNγ in the secretomes were quantified by ELISA using the human IFNγ ELISA KIT (Sigma-Merck, Madrid, Spain), following the manufacturer’s instructions.

### 2.6. Flow Cytometry

Human LPMCs were rinsed and dyed with fluorochrome-conjugated antibodies against CD45, CD14, CD64, CD206, CD16, CD86 and WNT2b (Appendix A). To exclude non-viable cells, we used a LIVE/DEADTM Fixable Near-IR Dead Cell SRAIN Kit (Thermo Fisher Scientific, Valencia, Spain). LSR Fortessa TM X-20 cytometer (BD Biosciences, Madrid, Spain) and DIVA software (BD Biosciences, Madrid, Spain) were used to analyse the samples. Macrophages were detected within single live CD45, CD64 and CD14 positive cells based on forward and side scatter properties, as previously described [16] (Appendix A).

### 2.7. Protein Extraction and Western Blot Analysis

Intestinal tissue, HT29 or U937 cells were homogenized, and total proteins were extracted with PhosphoSafe™ Extraction Reagent. Protein levels were analysed by WB [7] using specific antibodies (Appendix A). Protein bands were normalised to Glyceraldehyde 3-phosphate dehydrogenase (GAPDH). The densitometry of the bands was quantified using the software Image J.

### 2.8. RNA Isolation and Real-Time Quantitative PCR (RT-qPCR)

RNA extraction and RT-PCR were performed as previously described [7]. Real-time PCR was carried out in a thermocycler LightCycler (Roche Diagnostics, San Cugat, Spain). Specific oligonucleotides were designed according to the sequences (Appendix A). β-ATIN was employed as the housekeeping gene. 

### 2.9. Immunofluorescence

Immunofluorescence for VIMENTIN (Appendix A) was performed in HT29 cells, as previously described [7].

### 2.10. Statistical Analysis

Data were represented as mean ± S.E.M. and compared by analysis of variance (one way-ANOVA when *F* achieved *p* < 0.05) with a Newman–Keuls post-hoc correction for multiple comparisons or an unpaired t-test when appropriate (Graph-Pad Software 6.0, San Diego, CA, USA). A *p*-value < 0.05 was considered to be statistically significant. 

## 3. Results

### 3.1. IFNγ, SNAIL and WNT6 Are Increased in Intestinal Tissue from CD Patients

Total RNA was extracted from CD patients’ samples with a stenotic (B2 behaviour) or fistulous (B3 behaviour) pattern and from control donors and used to analyse the expression of different cytokines by qPCR. An mRNA analysis demonstrated that individual cytokines were significantly increased in CD patients with a stenotic or fistulous pattern when compared to control donors. IFNγ mRNA was increased in both patterns with respect to the mucosa of disease-free patients (Figure 1a). The expression of IL1β, IL6, IL10 and IL17-mRNA was significantly higher in the fistulized group than in the stenotic group. An ELISA was performed in order to determine the concentration of IFNγ in the tissue samples and revealed significant increases in both behaviours with respect to control samples (Figure 1b). Moreover, these increases in IFNγ were significantly higher in B3 patients than in B2 patients. A Westenr blot analysis revealed an elevated expression of the IFNγ receptor in both behaviours compared to controls (Figure 1c).

It has been demonstrated that intestinal tissue from CD patients displays higher protein expression levels of WNT2b, βCATENIN and FZD4, while there is a reduction of ECADHERIN in B3 patients, and that this pathway triggers the EMT process through the FZD4 receptor [7]. In light of this, we analysed the protein expression of WNT6, VIMENTIN and SNAIL in CD patients. Our Western blot analysis revealed an elevated expression of WNT6 and VIMENTIN in patients with both behaviours compared to controls. The transcript factor SNAIL was specifically elevated in B3 patients (Figure 2).

### 3.2. CD16, CD86 and WNT2b Positive Macrophages Are Increased in Intestinal Tissue from B3 CD Patients

We next analysed the expression of CD206 (alternatively activated macrophages), CD86 (classically activated macrophages) and CD16 (fibrosis-related expression [16,17,18]) in a cell suspension from whole intestinal tissue from controls and CD patients (LPMCs). Flow cytometry demonstrated that the percentage of macrophages detected as CD45, CD14 and CD64 positive cells was very low in control samples and was significantly increased in CD samples (Figure 3a). A high percentage of macrophages were CD206-positive cells in all the samples analysed. In contrast, a very low percentage of macrophages were CD86-positive cells or CD16-positive cells in control samples, the numbers of which were significantly increased in intestinal tissue from CD patients, reaching statistical significance in the fistulised group (B3) (Figure 3b). In line with these results, mRNA expression of CD206, CD86 and CD16 was enhanced in intestinal samples from CD compared with those from non-IBD patients. The expression of CD206 was similar among B2 and B3 CD patients, but the mRNA expression of both CD16 and CD86 was significantly higher in the fistulised group. Leukocyte adhesion markers did not differ among the three groups (Figure 3b).

With the aim of exploring whether the cytokine-rich tissue environment in CD patients was responsible for the CD86/CD16 expression observed, human MDMs cells were differentiated into macrophages in the presence of control or B2/B3 secretomes. The results revealed that the mRNA expression of CD86 and CD16 was significantly higher in cells treated with the B3 secretome than in those treated with the non-IBD or B2 secretomes. CD206 expression remained constant in all three groups (Figure 3c).

Macrophages secrete different kinds of ligands involved in the WNT pathway depending on their phenotype [12,13]. Accordingly, WNT2b expression was detected in around half the macrophages in control tissue, and this percentage was significantly increased in samples from B3-CD patients (Figure 4a). Analysis of the relationship between the expression of WNT2b and CD16 in the macrophage population indicated that the expression of WNT2b was analogous in CD16-positive and CD16-negative macrophages in control tissue. In samples from B2-CD patients, the expression of this ligand seemed to be reduced specifically in CD16-negative macrophages. Finally, most macrophages isolated from B3-CD samples expressed WNT2b irrespective of whether they were CD16-positive or CD16-negative (Figure 3b). In this way, the mRNA expression of *WNT2b* was significantly higher in human PBMC cells that had been differentiated into macrophages in the presence of the secretome from B3 intestinal tissue than in those treated with control or B2 secretomes (Figure 4c).

### 3.3. IFNγ-Treated Macrophages Exhibit Increased Expression of CD86, CD16 and WNT2b

In order to determine the expression of macrophage markers under the influence of the main cytokines present in the surgical resections from controls and B2- and B3-CD patients, we treated the U937 macrophage cell line with TNFα, IFNγ, IL1β, IL4 or IL10 for 4 days. The data showed that IFNγ was the only cytokine able to significantly increase CD86 and CD16 expression in U937 cells at both the mRNA and protein levels (Figure 5 and Appendix A).

We also analysed the expression of the known soluble EMT mediator TGFβ and of WNT2b and WNT6 ligands in macrophages treated with different cytokines. A protein analysis revealed that IFNγ-treated U937 cells significantly overexpressed WNT2b, while IL10-U937-treated cells significantly overexpressed WNT6 (Figure 6) and only IL1β-treated U937 cells exhibited increased TGFβ protein expression (Figure 6). The mRNA studies showed no significant differences between the genes analysed (Appendix A).

### 3.4. IFNγ-Treated Macrophages Activate the EMT Process in Colonic Epithelial Cells through the FZD4-Dependent WNT Pathway

To determine the influence of macrophages in the EMT process triggered by different cytokines, we co-cultured U937 macrophages—pretreated with TNFα, IFNγ, IL1β, IL4 or IL10 for 4 days—with HT29, a colonic epithelial cell line, for 3 days. The mRNA data showed that IFNγ-treated U937 macrophages increased some fibrosis markers in epithelial cells with respect to the vehicle (*VIMENTIN*, *αSMA*, *FSP1*, *SNAIL1*, *COLA1* or *COLA2*, *FGFR2* and *ENAH*) (Figure 7a). In line with this, Western blot analysis revealed a significant increase in VIMENTIN and SNAIL in HT29 cells co-cultured with IFNγ-treated U937 cells with respect to other co-cultures (Figure 7b). Immunofluorescence experiments revealed an increased expression of VIMENTIN in HT29 cells induced by IFNγ-treated U937 cells (Figure 7b) with respect to the vehicle. Analysis of the WNT pathway showed a significant increase in WNT2b and WNT6 ligands and βCATENIN stabilisation in HT29 cells co-cultured with IFNγ-treated U937 cells (Figure 7b).

In order to explore the role of the WNT pathway in the EMT process triggered by IFNγ-treated U937 macrophages, we pre-treated HT29 cells with the WNT inhibitor XAV939 or with the vector *miFZD4* prior to co-culture with IFNγ-treated U937 cells. The WB data revealed that XAV939 (Figure 8a) and *miFZD4* (Figure 8b) significantly blocked the increase in the expression of VIMENTIN and SNAIL and the accumulation of βCATENIN induced by IFNγ-treated U937 cells. Co-culture with IL1β, IL4 or IL10-U937 did not produce significant changes (Appendix A). 

## 4. Discussion

In this work, we postulate that macrophages present in an inflammatory context, such as B2 or B3 CD behaviours, are able to activate the EMT through activation of the WNT pathway.

The analyses of the intestinal tissue of CD patients who underwent surgery due to stricturing (B2 behaviour) or non-perineal fistulizing disease (B3 behaviour) revealed an increase in IFNγ and its receptor AF1, and a high number of CD16-positive macrophages. We also observed an accumulation of the WNT6 ligand, along with an increase in VIMENTIN expression, in both behaviours. Of interest, samples from patients with non-perineal fistulizing disease displayed a pronounced increase in IFNγ and CD86/CD16-positive macrophages with respect to those from intestines with stricturing behaviour, along with a significant expression of the transcription factor SNAIL. 

Both strictures and fistulae constitute important complications of CD. There is solid evidence to suggest that penetrating complications of this pathology do not occur in the absence of bowel strictures [19,20], but the mechanisms that favour fistula development over fibrosis are still unknown. In a previous study by our group, we demonstrated that EMT was more prevalent in intestinal tissue surrounding the fistula tract in B3 CD patients than in the stricture of B2 CD patients, which pointed to a specific interaction between WNT2b and FZD4 as a possible mechanism of EMT induction [7]. In the present work, our data led us to consider the possible role of IFNγ in the induction of EMT through CD16/CD86 macrophages since both these parameters were increased and accentuated in the intestinal tissue of B3 patients and were only partially altered in B2 patients. By means of an in vitro study, we show herein, for the first time, that CD86/CD16 macrophages under the influence of IFNγ increase the protein expression of VIMENTIN, an EMT protein, in epithelial cells. In addition, we demonstrate an increase in the number of members of the SNAIL transcription factor and accumulation of WNT2b, WNT6 and βCATENIN in epithelial cells under the same conditions. Interestingly, although we analysed the influence of other cytokines that are also prevalent in CD complications, only IFNγ-treated macrophages displayed a consistent response, which was partially reverted when epithelial cells were treated with XAV939 or *miFZD4*. ECADHERIN is reduced in B3 patients [7]; however, in our in vitro study, none of the treated macrophages significantly modified its expression in the epithelium, which suggests that other mechanisms must be involved. 

In the context of IBD, IFNγ has also been related to intestinal epithelial homeostasis through converging βCATENIN signalling pathways [21]. Our present data show that IFNγ alone did not trigger the EMT process in colonic cells (Appendix A), a function that was performed only by IFNγ-treated macrophages. This points to a soluble mediator released by macrophages as the intermediary of this process, and we believe WNT2b may be responsible. 

The WNT signalling pathway has been shown to play a fundamental role in the human lung [22] and kidney [23] fibrosis, but its role in fistula pathogenesis remains unclear. Our previous data revealed an increased accumulation of βCATENIN and FZD4 in both behaviours, but only WNT2b expression was up-regulated in non-perineal fistulizing tissue [7]. WNT2b is an activator of the WNT/βCATENIN pathway [24,25], and is produced in, but also outside, the epithelium [7,26]. In our in vitro study, WNT2b was up-regulated in the following two cell types: on the one hand, in HT29 colonic cells co-cultured with macrophages exposed to IFNγ; on the other hand, in macrophages exposed to IFNγ. Interestingly, WNT2b expression was detected in around half of the macrophages in our control tissue, a percentage that was significantly higher in fistulized samples. These cells accumulated in the intestine of CD patients irrespective of clinical behaviour, but we observed that the percentage of macrophages expressing WNT2b increased only in our fistulizing patients. In light of all this evidence, it is feasible to expect an accumulation of WNT2b of both macrophage and epithelial origin in CD patients with persistently high IFNγ levels.

In our in vitro study, IL1β and IFNγ induced the expression of CD86, but only the latter significantly stimulated the expression of CD16 and WNT2b, markers that have been implicated in IBD fibrosis [7,16]. CD16 cells also accumulated in the intestine of CD patients irrespective of clinical behaviour, but the percentage of macrophages expressing WNT2b increased only in our fistulized patients. Of note, human monocyte differentiation into macrophages in the presence of the secretome from B3-CD samples, which had a high IFNγ content, was associated with an increased expression of WNT2b and CD16, while the same was not observed with the secretome from B2-CD samples. These results lead us to suspect that CD16-positive macrophages are an extra source of WNT2b when penetrating behaviour is observed. In line with this, flow cytometry analysis revealed an extra pool of CD16-positive macrophages expressing WNT2b in intestine samples from B3-CD patients. However, expression of the ligand was also detected in CD16-negative macrophages in the same tissue, in a similar manner to that observed in control samples. It seems that most of the macrophages present in the intestinal tissue surrounding the fistula tract express WNT2b, and that this is not the case in the stricture of B2 CD patients. It is known that the tissue microenvironment defines macrophage phenotype, and our results, in line with previous studies [19,27], demonstrate a higher mononuclear inflammation and expression of pro-inflammatory cytokines, such as IFNγ, in the intestine of fistulizing vs. stricturing patients.

IFNγ, together with lipopolysaccharide, is routinely used to differentiate in vitro macrophages towards a pro-inflammatory (M1) phenotype, and M1 macrophages are characterised by the expression of surface markers such as CD86 or iNOs. On the other hand, previous studies have shown that levels of CD16+ circulating monocytes and mucosa-infiltrating cells increase with disease activity [28,29,30] and are related to fibrosis in IBD [16]. In this sense, our data suggest that an IFNγ-rich inflammatory outburst, especially in the context of fistulizing disease, increases the CD86/CD16 subset, is responsible for activating the EMT process, and contributes to intestinal fibrosis in CD. Considering the available evidence, controlling IFNγ levels in the mucosa of CD patients may be a therapeutic tool to prevent future complications. Indeed, IFNγ has emerged as a therapeutic target in several autoimmune diseases [31], and the present work highlights the possibility of employing IFNγ as a therapeutic target in immune diseases.

As a whole, our study demonstrates that IFNγ-treated macrophages increase epithelial mesenquimal transition through activation of the WNT pathway. We highlight the possible role of IFNγ-treated macrophages in the complications related to Crohn’s disease. A better characterization of the role played by IFNγ-treated macrophages in this process may help to develop new therapeutic approaches to prevent CD complications.

## Figures and Tables

**Figure 1 biomedicines-10-01093-f001:**
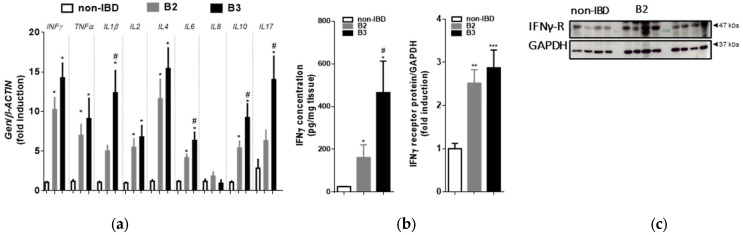
Changes in CD (Crohn’s Disease) intestinal tissue cytokines: (**a**) Surgical resections from control and B2- and B3-CD patients were frozen for qRT-PCR analysis. Graph shows quantification of the mRNA expression of cytokines. Data correspond to relative values vs. expression of the housekeeping gene, and are calculated as fold induction vs. the non-IBD (non- Inflammatory Bowel Disease) group (*n* = 10 per group); (**b**) Secretomes from B2 (*n* = 10) and B3 (*n* = 10) patients and control donors (*n* = 10) were evaluated for the presence of IFNγ by ELISA. The data presented are the mean IFNγ concentrations ± SEM in CD patients and control donors; (**c**) Graph shows protein expression of the IFNγ receptor (AF-1). Images correspond to a Western blot representation of 4 patients in each group (*n* = 12 per group). In all cases, bars represent mean ± SEM. Significant differences vs. the non-IBD group are shown by * *p* < 0.05, ** *p* < 0.01 or *** *p* < 0.001 and vs. the B2-CD samples by # *p* < 0.05.

**Figure 2 biomedicines-10-01093-f002:**
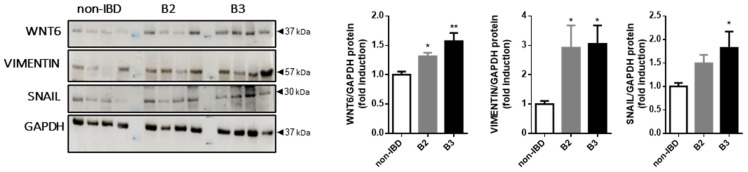
Changes in the WNT6 ligand, VIMENTIN and SNAIL in CD intestinal tissue: Graphs show protein expression of WNT6 (*n* = 8), VIMENTIN (*n* = 8) and SNAIL (*n* = 8). Images correspond to a Western blot representation of 4 different patients in each group. Significant differences vs. the non-IBD group are shown by * *p* < 0.05 or ** *p* < 0.01.

**Figure 3 biomedicines-10-01093-f003:**
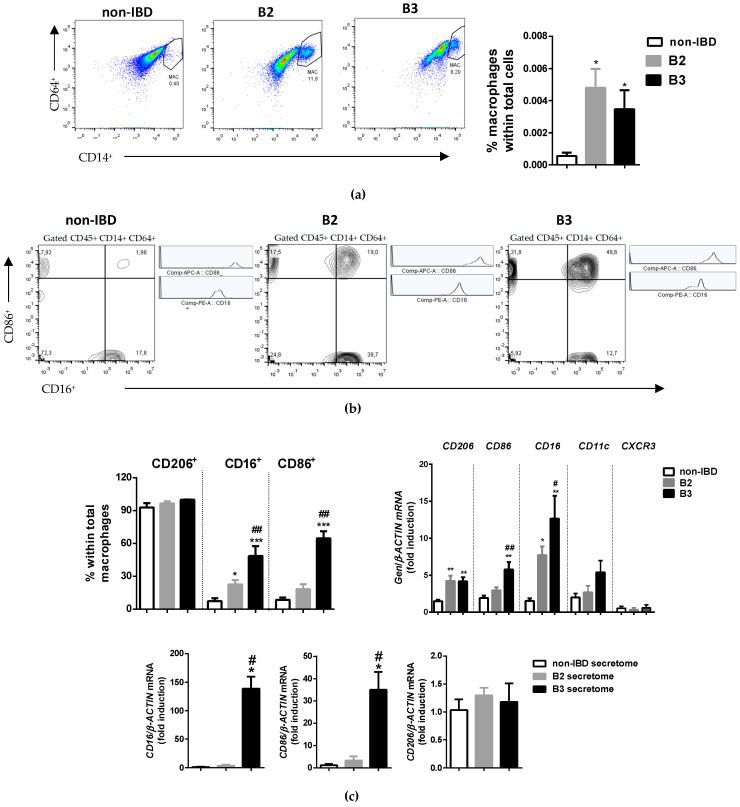
CD16- and CD86-positive macrophages are increased in intestinal tissue from B3 CD patients. LPMCs obtained from surgical resections were stained with CD45, CD14 and CD64 antibodies and identified by flow cytometry: (**a**) Lamina propria macrophages were identified as single live CD45+ CD14+ CD64+ cells (Appendix A); representative contour plots show macrophages in LPMCs cells from the intestine and the graph shows quantification of these cells (*n* ≥ 7 per experimental group); (**b**) Representative contour plots and histograms showing CD16- and CD86- positive cells in CD45+ CD14+ CD64+ lineage, graph represents the percentage of CD206-, CD16- and CD86-positive cells within total macrophages in intestinal tissue and the mRNA expression of these markers in intestinal tissue (*n* ≥ 7 per experimental group); (**c**) Human MDMs cells were differentiated into macrophages in the presence of secretomes from non-IBD or B2/B3 ileal tissue. The expression of CD16, CD86 and CD206 was analysed by RT-PCR (*n* = 4 per experimental group). In all cases, the bars in the graphs represent mean ± SEM. Significant differences vs. the non-IBD group are shown by * *p* < 0.05, ** *p* < 0.01 or *** *p* < 0.001 and vs. the B2 samples by # *p* < 0.05 or ## *p* < 0.01.

**Figure 4 biomedicines-10-01093-f004:**
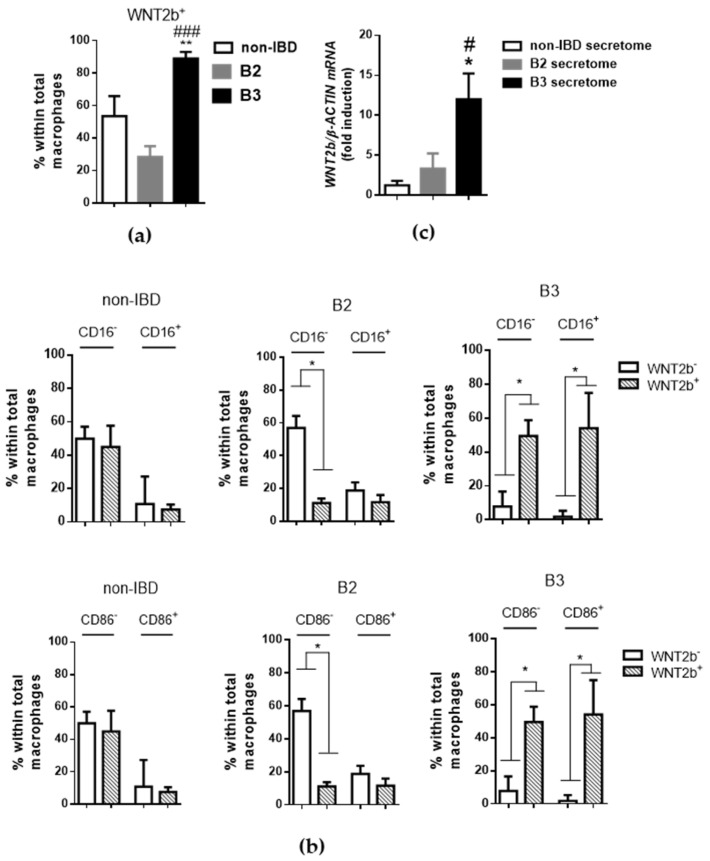
CD16/CD86-positive macrophages expressing WNT2b are increased in intestinal tissue from B3 CD patients. LPMCs obtained from surgical resections were stained with CD45, CD14, CD64 and WNT2b antibodies and identified by flow cytometry: (**a**) Graph represents the percentage of WNT2b-positive cells within total macrophages in intestinal tissue (*n* ≥ 6 per experimental group); (**b**) Graphs showing the percentage of CD16/WNT2b- and CD86/WNT2b-positive cells or their distribution within total macrophages (*n* ≥ 6 per experimental group); (**c**) Human MDMs cells were differentiated into macrophages in the presence of secretomes from non-IBD or B2/B3 ileal tissue. The expression of WNT2b was analysed by RT-PCR (*n* = 3 per experimental group). In all cases, the bars in the graphs represent mean ± SEM, and significant differences vs. the non-IBD group are shown by * *p* < 0.05 or ** *p* < 0.01 and vs. B2 CD samples by # *p* < 0.05 or ### *p* < 0.001.

**Figure 5 biomedicines-10-01093-f005:**
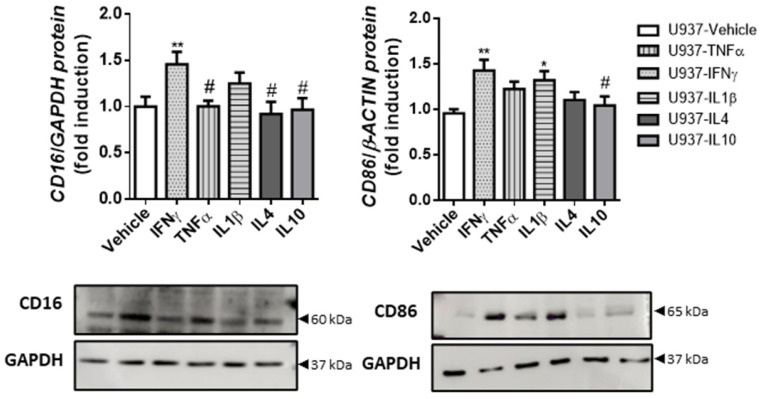
IFNγ-treated macrophages exhibit increased CD86 and CD16 expression. U937 cells were treated with IFNγ, TNFα, IL1β, IL4 or IL10, or vehicle for 4 days. Graphs show the protein expression of CD16 (*n* = 11) and CD86 (*n* = 20). The images correspond to a representative Western blot. In all cases, the bars in the graphs represent mean ± SEM, and significant differences vs. U937-vehicle cells are shown by * *p* < 0.05 and ** *p* < 0.01, and vs. U937-IFNγ by # *p* < 0.05.

**Figure 6 biomedicines-10-01093-f006:**
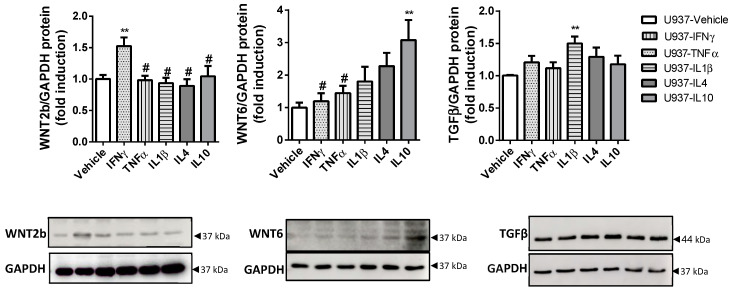
IFNγ-treated macrophages exhibit increased WNT2b expression. U937 cells were treated with IFNγ, TNFα, IL1β, IL4 or IL10, or vehicle for 4 days: Graphs show the protein expression of WNT2b (*n* = 8), WNT6 (*n* = 8) and TGFβ (*n* = 4). The images correspond to a representative Western blot. In all cases, the bars in the graphs represent mean ± SEM, and significant differences vs. U937-vehicle cells are shown by ** *p* < 0.01, and vs. U937-IFNγ by # *p* < 0.05.

**Figure 7 biomedicines-10-01093-f007:**
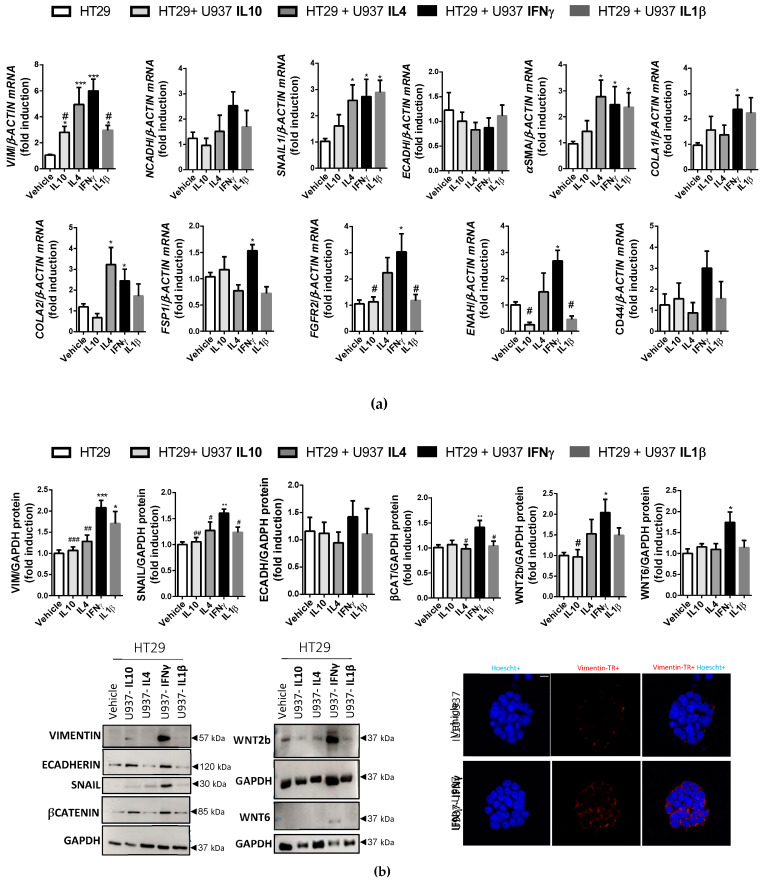
IFNγ-treated macrophages activate the EMT (Epithelial Mesenchymal Transition) process in colonic epithelial cells and the WNT pathway. HT29 cells were co-cultured for 3 days with U937 cells pre-treated with IFNγ, IL1β, IL4, IL10 or vehicle for 4 days: (**a**) Graphs show the mRNA expression of fibrosis markers (*n* ≥ 5 per experimental group); (**b**) Graphs show the protein expression of EMT markers and WNT pathway components. Images correspond to a representative Western blot (*n* ≥ 4 per experimental group). Representative images showing nuclear staining (Hoechst) and VIMENTIN (Tex Red) in HT29 cells treated with vehicle or IFNγ-U937 for 3 days (scale bar 100 μm). In all cases, the bars in the graphs represent mean ± SEM, and significant differences vs. HT29-vehicle cells are shown by * *p* < 0.05, ** *p* < 0.01 or *** *p* < 0.01 and vs. HT29-U937/IFNγ-treated U937 cells are shown by # *p* < 0.05, ## *p* < 0.01 or ### *p* < 0.001.

**Figure 8 biomedicines-10-01093-f008:**
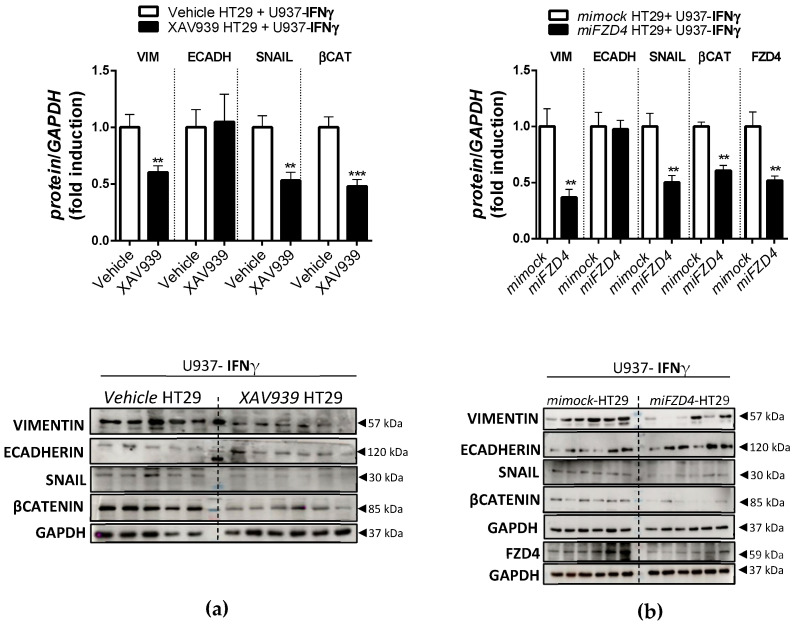
HT29 treatment with XAV939 or *miFZD4* inhibits the EMT process initiated by treating U937 cells with IFNγ. HT29 cells were treated with XAV939 or *miFZD4* and then co-cultured for 3 days with U937 cells that had been pre-treated with IFNγ for 4 days: (**a**) Graphs show the protein expression of EMT markers and βCATENIN in HT29 treated with XAV939 during the co-culture with U937-IFNγ cells. The images correspond to a representative Western blot (*n* ≥ 11 per experimental group). The bars in the graphs represent mean ± SEM, and significant differences vs. HT29-IFNγ-treated U937 cells are shown by ** *p* < 0.01 and *** *p* < 0.001. (**b**) Graphs show the protein expression of EMT markers and βCATENIN in HT29 cells pre-treated with *miFZD4* or *mock* vectors and subsequently co-cultured with U937 cells. Images correspond to a representative Western blot (*n* = 12 per experimental group). Bars in graphs represent mean ± SEM, and significant differences vs. HT29-mock cells are shown by ** *p* < 0.01.

**Table 1 biomedicines-10-01093-t001:** Information regarding CD (Crohn’s Disease) and non-IBD (non- Inflammatory Bowel Disease) patients.

		CD B2	CD B3	Non-IBD
**Number of Patients**		23	18	15
Age	Median	44	43	64
	Interval	[18–77]	[15–82]	[41–89]
Gender	Male	9	7	7
	Female	14	11	8
**Concomitant Medication**				
Azathioprine		18	8	-
Methotrexate		5	-	-
6-Mercaptopurine		-	3	-
Biological Therapy (anti-TNFα)		23	18	-

## Data Availability

The data presented in this study are available on request from the corresponding author.

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
