# Peer review of "IFNγ-Treated Macrophages Induce EMT through the WNT Pathway: Relevance in Crohn’s Disease"

_biomedicines, 2022, doi:10.3390/biomedicines10051093_

Round 1
Reviewer 1 Report
In this work Macias-Ceja et al. analyse the role played by macrophages in the EMT process related to fibrosis in Crohn’s disease. This study is very interesting and well written. There are some main points that need to addressed. Here are some suggestions:
- A high throughput analysis of the B2 and B3 secretome employed for the human mononuclear cells differentiation into macrophages will provide more insights into the factors or combination of factors responsible for the phenotype and markers expressed. It is important to show the representative images of the flow cytometry showing the expression of these markers (Figure 3c)
- IFN-g is known to promote EMT in the tumor microenvironment. That IFNg can induce EMT is already known. How IFNg-induced EMT can be correlated to the complex Crohn's disease context is needed. It is important to support the EMT-gene expression analysis on HT-29 cells with histological findings, for instance, morphological changes documented by microscopic images, or cellular motility, in the present context. Can EMT-associated events be observed in patient samples,too?
- Is EMT induced by IFN-g partial or complete? E-cadherin expression is not downregulated. A thorough analysis of EMT typing should be performed to clarify this point, for example by analyzing EMT-related gene isoform switches that occur during EMT (example, ESRP1 expression and its targets such as FGFR2, CD44 or ENAH by RT-PCR).
- Do the B2 and B3 also induce EMT in the HT29 cells?
- Can the authors interpret the results of the IFNg treatment in terms of M1/M2 macrophage phenotype? Whether IFNg can skew the macrophages towards an M1 rather than M2 phenotype?
Author Response
Dear Editor,
Following your instructions, we are resubmitting our manuscript referenced 1658882 and entitled “IFNγ-treated macrophages induce EMT through the WNT pathway: relevance in Crohn´s disease “ to Biomedicines.
Below we have answered point-by-point all the questions and queries raised by reviewers.
Reviewer 1:
In this work Macias-Ceja et al. analyse the role played by macrophages in the EMT process related to fibrosis in Crohn’s disease. This study is very interesting and well written. There are some main points that need to addressed. Here are some suggestions:
- A high throughput analysis of the B2 and B3 secretome employed for the human mononuclear cells differentiation into macrophages will provide more insights into the factors or combination of factors responsible for the phenotype and markers expressed. It is important to show the representative images of the flow cytometry showing the expression of these markers (Figure 3c)
- Answer: We appreciate the reviewer’s suggestion, and we agree it would be interesting to use ELISA to analyze the different cytokines present in the secretome. We will consider doing this in future projects (performing Multiplex immunoassays in the secretome), as it would be too complex to carry out within 10 days, which is the deadline we have been given for replying to the journal. In the present work we focus only on IFNγ because it is the only cytokine related to the EMT process described in our experiments, and because it is related with macrophage phenotype differentiation. Using ELISA, we analysed the concentration of IFNγ in the secretome, and its gene and receptor expression in human samples. With respect to the second question, the graphs in Figure 3c show the mRNA expression of CD16, CD86 and CD206 in macrophages treated with different secretomes; they do not represent flow cytometry. We realise now that the figure caption was confusing, and we have corrected it to clearly differentiate between cytometry and RT-PCR procedures. Furthermore, we have added representative images of the flow cytometry procedures to figure 3b.
- IFN-g is known to promote EMT in the tumor microenvironment. That IFNg can induce EMT is already known. How IFNg-induced EMT can be correlated to the complex Crohn's disease context is needed. It is important to support the EMT-gene expression analysis on HT-29 cells with histological findings, for instance, morphological changes documented by microscopic images, or cellular motility, in the present context. Can EMT-associated events be observed in patient samples,too?
- Answer: We appreciate the reviewer’s suggestion, and we have improved figure 7 by incorporating the results of immunofluorescence in HT29 cells co-cultured with IFNγ-U937 cells. However, in three days we have only observed the increased expression of VIMENTIN in HT29 cells, while the morphology was not modified significantly (Figure 7b). We feel that, ideally, experiments should be performed in a non-cancerous cell line, such as the HIEC-6 line (which we hope to obtain) or by incubating the co-culture for longer periods of time. Naturally, this would require new experiments with new protocols not included in the present work. Regarding the last question, in a previous study by our group, we demonstrated that EMT was more prevalent in intestinal tissue surrounding the fistula tract in B3 CD patients than in the stricture of B2 CD patients (Ortiz-Masià, et al., 2020). In the work in question we described the co-localisation of the epithelial cell marker E-CADHERIN with the mesenchymal cell marker VIMENTIN in crypts from intestinal penetrating CD patients (B3). In addition, RT-PCR analyses of isolated intestinal crypts obtained from B3-CD patients showed higher levels of the mesenchymal markers VIMENTIN, FSP1, and N-CADHERIN than those detected in cells from B2-CD patients. These observations, together with the fact that B3 samples also displayed a pronounced increase in IFNγ and CD86/CD16 macrophages with respect to those from intestines with stricturing behavior (B2), point to the relevance of IFNγ, mediated through CD86/CD16 macrophages, in the activation of the EMT process in CD.
- Is EMT induced by IFN-g partial or complete? E-cadherin expression is not downregulated. A thorough analysis of EMT typing should be performed to clarify this point, for example by analyzing EMT-related gene isoform switches that occur during EMT (example, ESRP1 expression and its targets such as FGFR2, CD44 or ENAH by RT-PCR).
- Answer: We appreciate the reviewer’s suggestion and have improved Figure 7a accordingly; it now shows the targets FGFR2, CD44 and ENAH by RT-PCR. The results reveal a significant increase in the mRNA of FGFR2 and ENAH in the HT29-IFNγ-U937 group with respect to the vehicle group.
- Do the B2 and B3 also induce EMT in the HT29 cells?
- Answer: We appreciate the reviewer’s suggestion and will bear it in mind for future projects, as we cannot perform such an experiment within the 10 days we have been given to reply, since incubation times exceed 10 days and secretome samples can only be obtained from patients in the operating room, the schedule of which is beyond our control.
- Can the authors interpret the results of the IFNg treatment in terms of M1/M2 macrophage phenotype? Whether IFNg can skew the macrophages towards an M1 rather than M2 phenotype?
-
- Answer: We appreciate the reviewer’s suggestion and we now comment on the role of IFNγ in the differentiation of macrophages towards M1 in the discussion section.
Reviewer 2 Report
The study has a potential interest. The authors indicate that IFNγ-rich microenvironment polarizes macrophages, inducing Epithelial mesenchymal transition through the WNT pathway.
The introduction section is informative and the objectives are clearly presented. It was reported in the this section the crucila role of WNT signaling pathway plays in EMT, and relationship between increased EMT and enhancing WNT2b/FZD4 in intestinal tissue from CD patients . This point is very interesting in severe cases of CD patients
The section Material end Methods is well conducted. More explanation is required about non IBD patients and biological therapies is anti-TNF alpha?(Table 1)
In Section Results, the authors reported thatI the mRNA expression of WNT2b was significantly higher in human PBMC cells that had been differentiated into macrophages in t
presence of the secretome from B3 intestinal tissue than in those treated with control or B2 secretomes (Figure 4c). Have identify the compounds of secretome?The results are good argued . I woul like to know your vision about a potential use anti-IFN-gamma or Il-4 in therapy of patients as preventive treatment.
I suggest a minor revision
Author Response
Dear Editor,
Following your instructions, we are resubmitting our manuscript referenced 1658882 and entitled “IFNγ-treated macrophages induce EMT through the WNT pathway: relevance in Crohn´s disease “ to Biomedicines.
Below we have answered point-by-point all the questions and queries raised by reviewers.
Reviewer 2:
Reviewer’s comment:
The study has a potential interest. The authors indicate that IFNγ-rich microenvironment polarizes macrophages, inducing Epithelial mesenchymal transition through the WNT pathway.
The introduction section is informative and the objectives are clearly presented. It was reported in the this section the crucila role of WNT signaling pathway plays in EMT, and relationship between increased EMT and enhancing WNT2b/FZD4 in intestinal tissue from CD patients . This point is very interesting in severe cases of CD patients
- The section Material end Methods is well conducted. More explanation is required about non IBD patients and biological therapies is anti-TNF alpha?(Table 1)
- Answer: We appreciate the reviewer’s suggestion and we now mention in the Material and Methods section that all CD patients had been prescribed anti-TNF treatment and underwent surgery due to complications. We have also stated that non-IBD samples were surgical specimens from unaffected mucosa from the terminal ileum of right colon cancer patients and that non-IBD patients were not undergoing chemotherapy before or at the time of surgery.
- In Section Results, the authors reported thatI the mRNA expression of WNT2b was significantly higher in human PBMC cells that had been differentiated into macrophages in the presence of the secretome from B3 intestinal tissue than in those treated with control or B2 secretomes (Figure 4c). Have identify the compounds of secretome?The results are good argued . I woul like to know your vision about a potential use anti-IFN-gamma or Il-4 in therapy of patients as preventive treatment.
- Answer: We appreciate the reviewer’s suggestion, and we agree it would be interesting to use ELISA to analyze the different cytokines present in the secretome. We will consider doing this in future projects (performing Multiplex immunoassays in the secretome), as it would be too complex to carry out within 10 days, which is the deadline we have been given for replying to the journal. In the present work we focus only on IFNγ because it is the only cytokine related to the EMT process described in our work, and because it is related with macrophage phenotype differentiation. We now include in the discussion a description of the potential use of anti-IFN as a preventive treatment.
I suggest a minor revision
Round 2
Reviewer 1 Report
The authors have answered to all comments from this Reviewer.
Of note, some images in the current version of the manuscript are overlapping.